# Accessing Voluntary HIV Testing in the Construction Industry: A Qualitative Analysis of Employee Interviews from the Test@Work Study

**DOI:** 10.3390/ijerph18084184

**Published:** 2021-04-15

**Authors:** Sarah Somerset, Catrin Evans, Holly Blake

**Affiliations:** 1School of Health Sciences, University of Nottingham, Nottingham NG7 2HA, UK; sarah.somerset@nottingham.ac.uk (S.S.); catrin.evans@nottingham.ac.uk (C.E.); 2NIHR Nottingham Biomedical Research Centre, Nottingham NG7 2UH, UK

**Keywords:** workplace, construction, men’s health, HIV, sexual health, health screening, health promotion

## Abstract

HIV, globally, remains a significant public health issue and community HIV testing can help to identify those with HIV at an early stage of disease. The workplace offers a prime location for provision of opt-in HIV testing as part of wider health promotion initiatives. The construction industry offers a key opportunity for HIV testing provision in a generally male-dominated group exhibiting some risky behaviors related to HIV. The intervention was an optional one-off individual health check with tailored health advice and signposting, offered to the construction workforce in health check events delivered as part of a large-scale multi-site research program called Test@Work. The events were undertaken at 10 participating organizations (21 events across 16 different sites), none had previously offered sexual health awareness or HIV testing to their workforce. Participants were invited to participate in a semi-structured interview following general health checks which included HIV testing. Out of 426 employees attending the health check events, 338 (79.3%) consented to interview on exit. Accessing HIV testing at work was valued because it was convenient, quick, and compatible with work demands. Interviewees identified HIV risks for construction including drug use, high numbers of sexual partners and job-related exposures, e.g., to used needles. Health seeking in construction was limited by stigma and low support, with particular barriers for non-permanent workers. The organization of the construction industry is complex with multiple organizations of different sizes having responsibility for varying numbers of employees. A disparity between organizational policies and employment circumstances is evident, and this generates significant health inequalities. To combat this, we recommend that organizations in the construction sector offer their employees awareness-raising around health behaviors and health protection in packages, such as toolbox talks. We recommend these be accompanied by annual health checks, including sexual health awareness and opt-in workplace HIV testing. This approach is highly acceptable to the workforce in the industry and removes barriers to access to healthcare.

## 1. Introduction

More than thirty years into the pandemic, HIV globally continues to be a significant public health issue with approximately 38 million people infected. Although great progress has been made in all aspects of HIV prevention and treatment, significant challenges remain [1]. One such challenge is the promotion of HIV testing in order to reduce numbers of those living with HIV who are undiagnosed and, thus, not accessing treatment (currently estimated as 19% of the global total). Similar challenges exist in the United Kingdom (UK). In the UK, in 2018, there were estimated to be 7500 people with undiagnosed HIV infections [2]. Of these, 4500 individuals lived outside of the London area and one of the highest rates was amongst white men who acquired HIV heterosexually (59% estimated as undiagnosed) [2]. Undiagnosed individuals risk transmitting HIV to others and developing more serious illness themselves. UK figures show that mainstream clinical services are not fully meeting the need for HIV testing. For example, within the country’s sexual health services, 760,031 (61%) of eligible attendees declined or were not offered an HIV test [2]. In order to address services and access gaps, UK public health policy recommends greater availability of HIV testing within the community setting, often referred to as ‘non-traditional’ (non-clinical) settings [2]. A key policy imperative, therefore, is to identify other sectors and settings where HIV awareness and testing may be offered. 

The workplace is a strategically important platform for promoting public health [3]. Prior research demonstrates the viability of the workplace setting for delivery of health promotion [4,5,6] and health screening [7,8,9]. However, in the United Kingdom (UK), sexual health is rarely included in workplace health programs [10] and the inclusion of HIV testing in workplace health promotion initiatives remains highly innovative. The inclusion of HIV awareness and testing in the workplace setting has been more commonly observed in international studies [11,12,13,14,15] and been found to reduce stigma around HIV and testing [11]. However, to our knowledge, only one UK study has reported on the inclusion of HIV in workplace health checks and found this approach to be well received by employers [10] and employees [16]. In the latter study, 92% of employees perceived workplace HIV testing to be acceptable, in a sample including low-waged, male, socially mobile, and migrant workers [16]. 

Workplace health promotion programs have potential to reach populations at risk who may not otherwise receive exposure to health promotion initiatives [17]. Few studies have investigated health promotion initiatives specifically in male-dominated industries. However, a higher prevalence of risky health behaviors is often found in industries dominated by men [18,19]. One example is the construction industry, in which >90% of the workforce in the UK identifies as male [20] and in which studies have shown high levels of HIV risk behavior, such as drug use and unprotected sex [12,21,22,23]. Furthermore, certain occupational factors often observed in the construction industry (such as high job strain, low social support and overtime work) have previously been associated with risk of ill-health (e.g., cardiovascular disease [24,25]). Personal and environmental factors are known to influence health behaviors [26], and temporary working arrangements coupled with work mobility often observed in male-dominated work populations can generate particular health concerns for men and their families [27]. Previous research has shown workforce benefits of health and lifestyle intervention in the construction industry (e.g., Reference [28]: physical activity, diet, and smoking), but workplace health interventions in the UK construction industry have not yet included voluntary HIV testing. This study is, therefore, highly innovative and timely, providing novel insights into real-world workplace health promotion and HIV testing with global relevance. The aim of this multi-site study (referred to as the ‘Test@Work’ program) was to investigate employees’ experiences of voluntary workplace HIV testing delivered as part of general health checks in the construction industry.

## 2. Materials and Methods

### 2.1. Study Design

To address the aim of this study, a qualitative descriptive approach was adopted to explore employee’s perceptions and experiences of HIV testing in the workplace [29,30]. This approach draws on an interpretive philosophical orientation and seeks to present a ‘coherent conceptual description’ of a phenomenon of interest, usually expressed in terms of key themes that account for similarities and variations in the data. Ethical approval for the study was received from the University of Nottingham Faculty of Medicine and Health Sciences Research Ethics Committee, Ref: LT20042016). Qualitative semi-structured interviews took place in the Midlands area of the UK between August 2019 and March 2020. This study adheres to the consolidated criteria for reporting qualitative studies (COREQ) guidelines [31] (Appendix A).

### 2.2. Workplace Health Checks

The intervention was an optional one-off individual health check with tailored health advice and signposting, offered to the construction workforce in health check events delivered as part of the Test@Work program [32]. The events were undertaken at 10 participating organizations across 16 construction sites. None of the organizations recruited had previously offered sexual health awareness or HIV testing to their workforce. The project researcher arranged events with organizations and agreed on a mutual date and time to host a health check event. The organizations advertised the health check event on their site and employees were allocated a slot on an appointment basis. On arrival, participants were asked to complete a short questionnaire and were given a pack containing additional information about the health checks. The health checks took place in a convenient space identified by the participating organizations. The general health checks took place in one room, and, in line with clinical governance requirements, a separate private room was provided for HIV testing and consultations. The health checks included a range of optional checks, including height, weight, body mass index, waist-to-hip ratio, blood pressure, mental health screening, and an Alere Determine™ HIV-1/2 test (Alere Scarborough, Inc., North Chicago, IL, US). Participants were provided with personal individualized feedback on their health check results; they received tailored health advice and signposting and were provided with a take-away health resource pack. The rationale, content, and delivery of the health checks and health advice is described in detail elsewhere, together with further details about the delivery team and their training [32]. Line managers at the participating organizations were provided with an evidence-based digital toolkit about the rationale for workplace health promotion and health screening, with details about the processes for HIV testing [33] to support the promotion of the event to their staff. 

### 2.3. Participants 

All employees of the participating organizations were invited to take part in the health checks. ‘Employees’ in this context refers to anyone associated with the organization and working on the construction site on the day of the health check event. This includes individuals employed by the participating organizations, as well as contractors and agency workers. Participants in this study were employees who had taken part in a workplace health check at one of the 16 sites and consented to take part in an interview.

### 2.4. Procedure

All participants who took part in a workplace health check were invited to take part in a brief exit-interview immediately after their health check (a convenience sampling approach). All participants received an information sheet about the study and provided written informed consent. A semi-structured interview guide was developed by the project team (Appendix A). The aim of the interviews was to gain insight into participants’ prior experience of health checks in the workplace, their views towards the health check events and specifically their views towards the inclusion of HIV testing as part of this.

The interviews were held during working hours, in a room at the participant’s worksite. The type of room varied according to the facilities at each site; but interviews were always held away from construction activity for safety reasons and in a separate area to the health check itself. Depending on the facilities available onsite, this was either in a separate room or in a suitable private space away from the health checks. Signage was used to prevent interruption of private and confidential discussions. The number of participants having health checks at each location depended on the size of the organization, and recruitment success at the sites which was determined by employee availability. The number of health check participants consenting to be interviewed immediately after the check primarily depended on their availability and competing work demands.

Participants completed a brief questionnaire providing demographic information including age, gender, country of birth, ethnicity, sexual orientation, employment type, whether they spoke English as first language, and whether they had taken an HIV test previously. They were also asked to rate how healthy they felt on the day of the event on a scale of 1 to 10, where 1 = poor health and 10 = excellent health. Interviews were audio-recorded with consent and transcribed verbatim.

### 2.5. Data Analysis

Transcripts were managed in NVivo qualitative data analysis software (QSR International Pty Ltd. Version 12, 2018, Chadstone, VIC, Australia) and analyzed using an inductive thematic analysis approach [34]. This allowed for the interpretation of themes from the data and for these to be grounded in the real-life experiences of the participants. To maximize analytical rigor, a team approach was utilized for data analysis. The project researcher (SS) reviewed transcripts and audio recordings to familiarize herself with the data prior to commencing formal analysis. The project researcher coded the data and developed an accompanying code book, organized into sub-themes and themes. The same process was undertaken independently by a second member of the research team (CE) with a sub-sample of 15 transcripts. Of a total 40 coded items, the researchers agreed on 32. The remaining eight were revised and incorporated into the other codes (*n* = 32). This represents an 80% agreement in the initial coding. The coding scheme and evolving themes were then compared and contrasted and discussed with the wider research team. A final framework was agreed and applied across the data set with new codes added where appropriate. 

## 3. Results

### 3.1. Sample Characteristics

A total of 464 employees were invited to participate in a health check, of these, 426 participated (men: *n* = 348, 81.7%; women *n* = 78, 18.3%) in the health check events, 28 men declined, and 10 men did not return after a fire drill on location. Of those participating, 338 (79.3%) consented to be interviewed on exit. For the 88 participants declining interview, 50 (11.7%) were too busy, 13 (3.1%) experienced communication barriers, and 25 (5.9%) did not give a specific reason. Interviews took place across a range of organizations (1 medium, *n* = 50–249 employees; 9 large, *n* ≥ 249 employees). A total of 21 health check events were carried out across 16 different sites, including nine building sites and seven office sites within the construction industry. 

Interview participants were aged 17 to 67 years (mean = 40, s.d. = 11.938), with the majority of participants falling between 17 and 50 years. Most of the participants reported that they were White British (*n* = 352, 82.6%), spoke English as a first language (*n* = 397, 93.2%), and identified as heterosexual (*n* = 415, 97.4%). Participant characteristics can be found in Table 1. 

Of the interview participants, 181 (42.5%) had participated in some form of health check in the workplace previously. Three-quarters of the sample reported that they had never taken an HIV test previously (*n* = 325, 76.3%) with just 23% (*n* = 98) having tested before and <1% (*n* = 3) declining to answer the question. Over 80% (*n* = 348, 283 M, 65 F) of the sample opted to take an HIV test, with 18.3% (*n* = 78) declining. 

### 3.2. Qualitative Interviews

The analysis produced three overarching themes: (a) valuing the opportunity for an HIV test (Figure 1), (b) perceptions of HIV risk (Figure 2), and (c), health seeking amongst men in the construction industry (Figure 3). 

The interviews demonstrated that the majority of participants valued the offer of HIV testing as part of the workplace health check event:


*I thought it was really good, I think it is not something that you would go and naturally ask for, can I have one of those tests so if it is offered more people are likely to say yes because it is a good thing to get checked. (White male, age 26)*



*Well I think a lot of people would be afraid to get HIV tests done so I think it is good that it is being promoted more and more people can do it, get themselves checked before it is too late, so it is a good idea. (White male, age 38)*


Thus, one of the main reasons for appreciating the HIV test was simply having the opportunity. In addition, individuals appreciated the ease of access, convenience of location, and noted a lack of awareness of alternative locations to access testing: 


*Just because… again just because it was available, I don’t feel I have any symptoms of it but erm you know I just thought why not, it’s here and I wouldn’t be sure where else to go? (White male, age 30)*


Many participants noted that those within the industry often travel long distances to attend construction sites across the country and do not have time to look after their health properly. As such, they felt that events like this were important as they facilitated the opportunity for employees to monitor their health, something they may not have ordinarily focused on: 


*Because you don’t have time with work hours…we don’t have time to go for check ups or see how you’re getting on at home so it is quite nice to be able to go and do it and not have to worry about not getting paid and everything else so that is really the reason why, make sure that I am healthy, and I am not too bad. (White male, age 48)*


Some participants indicated that they wanted to access this type of service but stated they would not have actively sought it out, primarily due to uncertainty as to how they would access an HIV test in other settings:


*I say I think it is good to have it...Because it is not like you are offered it at the doctors or anything, you have to go and seek it but not… it is not offered, do you know what I mean? (White male, age 39)*


Those opting to take a test generally did so because access was perceived to be easy, results were available quickly, and the consultation was private. Many of those participating in the health checks shared the time pressure they were under for job completion and expressed appreciation for the brevity of the health check and the fast-paced delivery of the HIV testing result (in general, this was available within 20 min of testing): 


*It was good, it was nice and quick, it doesn’t impact much on the day which for sites like this is important, I thought that was quite good, everything else I was happy with. (White male, age 34)*


Some participants noted that the availability of the HIV testing as part of the health check was unexpected. Despite provision of detailed information to the company representatives with whom the events were organized, some participants were, nonetheless, unaware of which checks would be on offer at the event until they arrived for their appointment. This led to some participants feeling ‘*alarmed*’ or ‘*surprised*’ that HIV testing was on offer but, nonetheless, saw its value once they had had an opportunity to think it over. The researcher explained all the checks were optional, and this provided reassurance to participants: 


*I think the range of options that have been offered have been very good and a bit surprised by the HIV test, I didn’t realize that was going to be there but while it is there, why not use it. (White male, age 57)*


Only a small minority of participants felt that the workplace setting was not appropriate for offering an HIV test due to concerns about privacy and confidentiality: 


*I do think it is a really good idea erm and the only thing I would say is the HIV checks I think is great as well erm however say if somebody did have it, it would be really awkward to walk out because you see by their facial expressions, they would be like oh god like nightmare so that might be awkward in an office-based environment I would say. (White female, age 20)*


Participants’ views on the relevance and value of the HIV test as part of a workplace health check were also influenced by their perception of HIV risk. This related to both personal lifestyle and to the workplace. Participants accessing the HIV testing perceived their level of risk in different ways. Some believed they were at low risk or held stereotypical views about risk categories, and some had declined the test for that reason:


*Yes, well I don’t know, I have always understood HIV is for gays and things like that do you know what I mean? I might be wrong but that is how I have always looked at it you know I am not gay why should I need an HIV? (White male, age 57)*


For others, it was an opportunity to check their health status irrespective of the level of perceived risk:


*Yes, I had never had it done but I have been with my wife for 10 years now so although I am not sleeping with multiple partners it is just always something which is good to know. (White male, age 36)*


Other employees reported that the decision to take a test was related to perceived potential risks within their own lifestyle or behavior. For example, some reported having unprotected sex with multiple partners, whether currently or prior to settling in their current relationships: 


*I love sex, with women erm… and without condoms so I needed the HIV test. (White male, age 28)*


Others reporting engaging in recreational drug use, and problem alcohol consumption, including heavy drinking or binge drinking, which could lead to unsafe sex: 


*I think it is positive, erm a lot of men on a building site are men, men who like to have a few drinks and might participate in recreational drugs erm… frequent sexual partners, so I think it is a very good thing for them to have that check. (Black male, age 52)*


When probed on their reasons not to have tested previously (for those that had not), the challenges of accessing testing were again referred to (where and how to do it), together with job-related factors, such as managing long working hours within the construction industry. 

Some participants perceived the workplace as a site of potential HIV risk and saw the test as an opportunity to assess their health status. This was most often related to fear of needle stick injury – for example when working in sewers or in houses that had been used by drug users:


*Just in general the HIV test, we do a lot of council work and we come across needles all around…So I had it [the HIV test] done just because I have never had that opportunity before. (White male, age 47)*


The majority of participants in this study were men. The interviews revealed that their discussions and views on HIV testing in the workplace were located within, and consistent with, broader patterns of health seeking behavior. The majority of participants reported a reluctance to seek healthcare assistance until they were symptomatic or unable to work as a result of a health issue, i.e., the participants exhibited *reactive* health seeking behavior rather than *proactive* health seeking behavior: 


*There is a lot of blokes out there that don’t do anything about it until they have to. I think you should know, really, you should know what sort of physical health you’re in, especially in the job we do. (White male, age 47)*



*Erm just because it was here, and I thought I never normally go to the doctors… I don’t need to go to the doctors so I don’t go, and I don’t really weigh myself or know anything about any health issues that I might have or anything. (White male, age 27)*


Two main reasons appeared to underlie this phenomenon – work related barriers and masculine culture. The work-related barriers included the aforementioned difficulties in accessing services due to varying locations of construction sites (the sample included many mobile workers), a perceived lack of appointment availability within general practice/primary care services (and an inability to book healthcare appointments in advance), long working hours and loss of pay if time was taken out of work to attend appointments: 


*You don’t have time with work hours and studies we don’t have time to go for check-ups or see how you’re getting on at home, so it is quite nice to be able to go and do it and not have to worry about not getting paid and everything else. (White male, age 35)*



*Due to the current economy social climate of Britain, it is tough for me to get in to my doctors…I would need to book time off work, and it would just become a bit stressful… well for me it is stressful so you know I just avoid the doctors like the plague to be honest with you. (White male, age 27)*


A perceived lack of support for health-seeking at work was another occupational barrier that was frequently alluded to by staff in all occupational roles within the industry, including managerial staff and site labourers. Employees reported a welcome recent increase in certain workplace schemes (particularly around mental health), but many still felt that more could be done:


*So we have got mental health first aiders available, we have got different safe call numbers like confidential numbers available for people to use, we do run days like this or like health MOT days, I suppose we could be better at it maybe by I don’t know buying things that they need or having initiatives or making them take breaks for exercise, I don’t know, even for myself because I work quite long hours and I travel a lot, I come from Walsall, erm yes so it is quite a long commute so yes. (White female, age 27)*


In particular, there appeared to be a ‘them and us’ attitude relating to the office staff and permanent staff members compared to contractors and agency staff. Permanent staff appeared to be more aware of resources they could access for assistance within the workplace. Conversely, those participants who were contractors and agency staff seemed less aware of what was available to them and discussed difficulties with accessing any services provided by the overarching companies that were running the building sites. They also perceived that there was a difference in the way they were treated by the organization in terms of taking time off for appointments: 


*Especially with this self-employment game it is like… you know you are kind of out here fending for yourself, so it is kind of good to you know just have that… because obviously you have got corporate companies that have people on the books and they tend to like look after their own… some places do, some places don’t but you know being self-employed you kind of… in the long grass on your own you know. (White male, age 32)*


Employees reported, in some instances, differences in the levels of support they may be offered when compared to permanent staff, especially given the regular movement between sites. There were also reports of variations in provisions offered to contract workers, such as access to free medical care, gym equipment, and various workplace schemes, that varied from site to site. It seemed that there was a lack of an industry wide standard for employee health and wellbeing across organizations and staff groups: 


*Erm - it depends for me because I work from site-to-site different erm contractors because I subcontract so it varies on the employer (White male, age 33)*


Employees also alluded to barriers associated with being in a male-dominated industry in which they perceived a stigma associated with men seeking healthcare advice:


*I think it is vital really because I think well especially being in construction it is a male*
*-dominated environment, I think a lot of men sort of have that stigma around attending a GP or somewhere to find out about their health in general. (White male, age 25)*



*I think more needs to be done by employers because I think particularly in the construction industry it is almost an attitude of man up or ship out type of thing and there is too much aggression in the workplace, I think. (White male, age 54)*


The bravado associated with being in a male-dominated environment was also linked to a perception, expressed by some, of bullying within the industry. Bullying was discussed in relation to work-related pressure and targets that are set within the construction industry, alongside work-related stress—all of which contributed to an atmosphere that was perceived as hostile to taking time off to seek healthcare: 


*There certainly is you know they treat us poorly and they will not get a lot out of us and guys are going off sick because the pressure…the bullying within the management filtering down to us…(White male, age 52)*



*Erm yes I think it [the health check] is good….but there is… a hypocrisy...is it ironic? I don’t know, that work is the biggest cause of stress I think for a lot of instances so I guess it is good that they are allowing us to do it – but working culture probably needs to change I think at the minute, in this industry it does, it is ridiculous. (South Asian male, age 40)*


Hence, employees reported that they were more likely to seek healthcare (or attend health checks) if they were allowed time away from their job by the employer to attend:


*Because my boss gave me time off to come in and do it, if not I probably wouldn’t have come. (White male, age 22)*


Taken together, these findings suggest that the workplace environment exerts a significant impact on the health seeking behavior of these individuals. Taking time out to attend health check-ups when not actually symptomatic (as is the case for an HIV test) was, thus, an uncommon and unsupported behavior. Many employees, however, expressed a desire for health check events to be provided by their employer or by external organizations an annual or biannual basis and felt that this would help them to monitor their health, as well as providing vital access to health-related information around lifestyle behaviors: 


*I think it is a good thing to do, it really is a good thing to do, we don’t do enough of it. It is the first one I have done here in 18 months, but my previous employer had done them a few times. Especially in construction where it is a male orientated industry, we and we’re not very good about health and going to the doctors, I think as men so… I am certainly not……and I think the HIV testing is a good thing as well because as you said a lot of people won’t be thinking that is even an issue when actually they could have it without even realizing so. (White female, age 26)*


## 4. Discussion

This is the first study to evaluate the perceptions of employees towards workplace health checks (specifically HIV testing) within the UK construction industry. The findings from this highly innovative study have world-wide relevance for addressing a global public health concern through health promotion and HIV testing in the workplace setting.

The study demonstrates that HIV testing, delivered in the context of a general health check, is highly acceptable to employees in this male-dominated sector and reached individuals who had never had an HIV test, as well as repeat testers. This is important since sexual health awareness and testing is rarely included within general health check provisions, particularly in the UK [7,10]. The majority of workers opted in for an HIV test which suggests that the workplace is a viable option for community delivery of HIV testing. This is in line with other research that advocates the workplace as an important platform for health promotion [3,35], and a key route to tackling issues with men’s health [36]. 

Many participants believed themselves to be at risk for HIV, mostly due to job-related health and safety concerns (e.g., risk of needle stick injuries), whereby workers felt they were regularly exposed to situations that were high risk, whether this was actual or perceived risk. Many participants also acknowledged a high prevalence of risky behaviors in the workforce (e.g., multiple sexual partners and/or recreational drug use). A smaller proportion of the sample opted out of the HIV test; these individuals generally perceived that they were not at risk for HIV, which could be related to reports of low-risk lifestyle choices, or potentially, misunderstandings about the way in which HIV is transmitted. 

The themes identified in this study offer novel insight into specific patterns of health seeking behaviors in this male-dominated, construction workforce. Engagement in workplace HIV testing was strongly related to the convenience of having a health check at work, making the most of the opportunity presented, ease of access, and a lack of awareness about other options for accessing testing services. The health checks were perceived to be successful primarily due to the availability of tests during the working day, rapidity of testing and feedback, and the way in which workplace testing broke down barriers to healthcare access for many of these employees. Reducing barriers to health services access is key, since it is already established that men access fewer primary care services than women [37] and are less likely to be proactive with regards to their health [38,39]. 

While there did not appear to be a perceived stigma around HIV testing in our sample per se, the normalization of testing within the workplace setting (a strategy advocated by prior researchers, e.g., Ishimaru et al, 2016 [40]), coupled with camaraderie between peers, seemed to increase uptake and reduce the stigma associated with men seeking healthcare advice. Participation in the workplace testing was facilitated by the perceived knowledge level and approachability of the testing service providers. 

The work environment within the construction industry, as described by participants (e.g., high workloads, long hours, and time pressures, insecure contracts, commuting, aggression, and masculinity), was uniformly perceived to be problematic and a disincentive to accessing health services. Job contracts and the pressures of financial insecurity were strong themes in this research and highly prevalent in the construction industry; this is relevant since there is generally a higher rate of temporary employment amongst men [27], and work mobility has previously been found to contribute to additional health concerns for men and their families [27]. 

### 4.1. Strengths and Limitations

This study provides novel and timely insights for workplace health promotion and HIV testing that have global relevance. Although a convenience sampling approach was used, the findings are drawn from a very large sample across all study sites. We were able to capture the views of most people that took part in the health checks. Since this was a male-dominated workforce, the number of women participants was small and albeit expected, we are not able to make meaningful gender comparisons. The interviews were relatively short due to the need for their completion during employee breaks; however, due to the large number of interviewees, we felt that data saturation was achieved [41,42,43]. We were unable to interview a small number of migrant workers who had a health check due to language barriers as translation was not available. The majority of participants were White British and may not be classed as a major risk population for HIV [2]; this may reflect the geographical regions in which the study was conducted, and replication elsewhere may generate a more diverse sample. However, our sample did include a high proportion of young men, who were highly mobile and many of whom reported engaging in high-risk behaviors. 

### 4.2. Recommendations for Future Research

Future studies may consider replicating this approach in other workplace contexts and settings, particularly in geographical regions with populations at high risk for HIV. The views of employers, migrant workers and women working in male-dominated environments could be further explored. Further in-depth process evaluation of workplace HIV testing intervention would add insights to this area, particularly around barriers to uptake and challenges to implementation. The positive views of construction workers towards HIV testing in the workplace indicates the potential value of promoting other areas of health via this industry, including other types of health screening, and promotion of both mental and physical health. There is potential to further explore approaches to reducing barriers to workplace health promotion relating to job contracts and the work environment in this sector. Finally, the COVID-19 pandemic has reduced access to healthcare services globally, and the value of workplace health interventions in this context could be further investigated. 

## 5. Conclusions

This is the first study to explore the perceptions of construction workers towards HIV awareness and testing in the workplace setting. Our large-scale, multi-site study demonstrates that working-age populations at risk for HIV can be reached outside of clinical settings and that workplace HIV testing is well-accepted in this male-dominated industry. This study shows the value of increasing access to testing through the workplace setting, and lessons learned about the nuances of this context have global relevance for addressing this significant public health concern. Construction is a complex industry, with overarching organizations often overseeing multiple smaller employers who have responsibilities for their own workforce. The disparity between organizational policies (or the lack of), and individual employment circumstances (contractual versus permanent) generates significant health inequalities and complicates the delivery of, and access to, employee health and wellbeing initiatives. There is a need for industry-wide policy on workforce health and wellbeing to reduce work-related health problems that have organizational impacts with regards to sickness absenteeism. There is also a clear need to advocate health protection behaviors in this workforce. Training is needed for line managers to emphasize the organizational value of workplace health and provide guidance around health screening at work. We recommend that organizations in this sector offer their employees awareness raising around health behavior and health protection (e.g., induction materials, health campaigns, and regular communications, such as ‘toolbox’ talks), together with annual or biannual general health checks. We advocate that, globally, workplace health interventions include sexual health awareness and opt-in workplace HIV testing, which is highly acceptable to the construction industry workforce and removes barriers to healthcare access.

## Figures and Tables

**Figure 1 ijerph-18-04184-f001:**
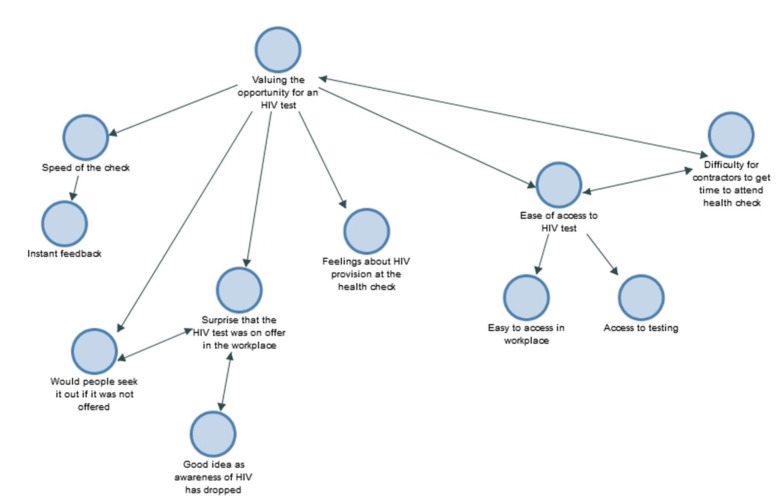
Theme (a): Valuing the opportunity for an HIV test.

**Figure 2 ijerph-18-04184-f002:**
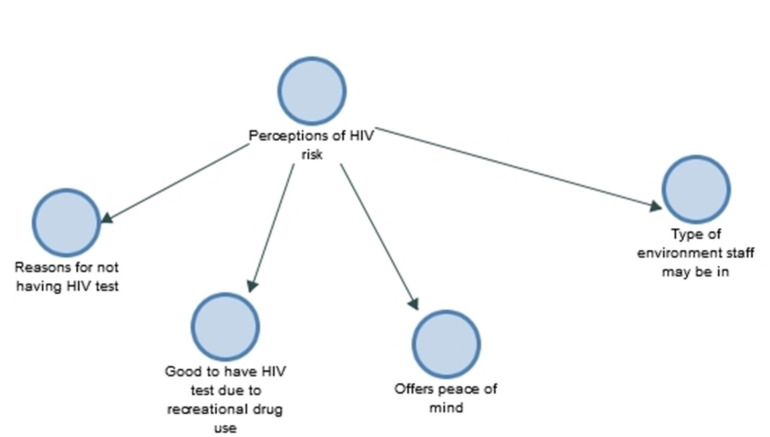
Theme (b): Perceptions of HIV risk.

**Figure 3 ijerph-18-04184-f003:**
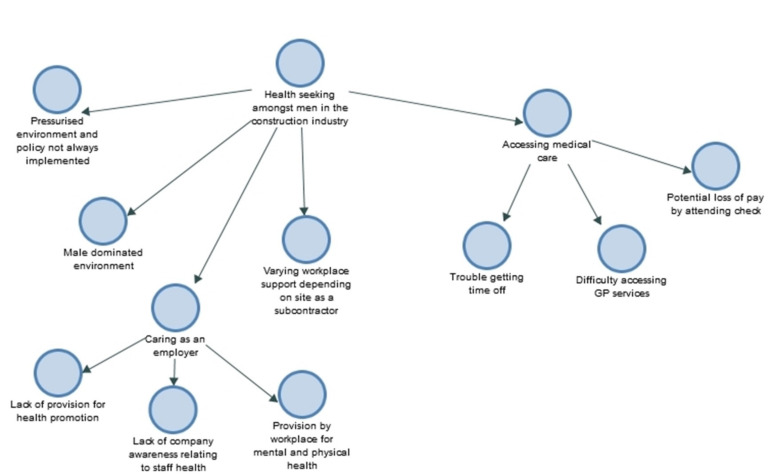
Theme (c): Health seeking amongst men in the construction industry.

**Table 1 ijerph-18-04184-t001:** Sample characteristics.

	Interview Participants ^†^ N = 426 (100%)		
	Men N = 348 (81.7 %)	Women N = 78 (18.3 %)	Total N = 426 (100%)
Age category (years) 17–30 31–40 41–50 51–60 61–70	90 (25.9) 98 (28.2) 86 (24.7) 55 (15.8) 19 (5.5)	16 (20.5) 23 (29.5) 18 (23.1) 18 (23.1) 3 (3.8)	106 (24.9) 121 (28.4) 104 (24.4) 73 (17.1) 22 (5.2)
English as first language Yes No Not stated	326 (93.7) 19 (5.5) 3 (0.9)	71 (91.0) 7 (9.0) -	397 (93.2) 26 (6.1) 3 (0.7)
Ethnicity British Irish Any other white background White and black Caribbean Any other mixed background Indian Pakistani Bangladeshi Caribbean African Other Ethnic groups Chinese Not stated	296 (85.1) 6 (1.7) 11 (3.2) 3 (0.9) 1 (0.3) - 18 (5.2) 2 (0.6) 2 (0.6) 6 (1.7) 1 (0.3) - 2 (0.6)	56 (71.8) - 6 (7.7) 1 (1.3) 1 (1.3) 1 (1.3) 5 (6.4) - - 5 (6.4) 1 (0.3) 1 (0.3) 1 (0.3)	352 (82.6) 6 (1.4) 17 (4.0) 4 (0.9) 2 (1.5) 1 (0.2) 23 (5.4) 2 (0.5) 2 (0.5) 11 (2.6) 2 (0.5) 1 (0.2) 3 (0.7)
Sexual orientation Heterosexual Homosexual Other Not stated	337 (96.8) 2 (0.6) 2 (0.6) 7 (2.0)	78 (100.0) - - -	415 (97.4) 2 (0.5) 2 (0.5) 7 (1.6)

^†^ No participants identified their gender as non-binary.

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
