# Peer review of "Accessing Voluntary HIV Testing in the Construction Industry: A Qualitative Analysis of Employee Interviews from the Test@Work Study"

_ijerph, 2021, doi:10.3390/ijerph18084184_

Round 1

Reviewer 1 Report

The manuscript investigates employees’ experience of voluntary workplace HIV testing which is a part of general health checkups in the construction industry which is a male-dominated sector and often involves high-risk behavior. The authors adopted a qualitative description approach for the study to understand employee’s mindset and their health check history including HIV testing. Based on the interviews, the authors recommended that employers should offer their employees more awareness about their health and offer regular health checkups including the option of HIV testing. The investigators provided enough background information and appropriate references. It was interesting to see authors include the real interview answers from the employees which gave us an idea into the mindset of the employees. I would have liked to see a few more interview answers from employees who did not think that HIV test at the workplace was a good idea. Construction being a male-dominated industry there were very few female interviews as the authors explained. Therefore, it would be fascinating to repeat the study in a female-dominated sector to see how different their mindset is. Overall, this was a very interesting study that provided immense insight into what is accepted in the workplace health checkup and that majority of the employees appreciate it along with HIV testing. The manuscript could be improved if the authors provided with future directions or if they plan to undertake any further study.

Author Response

Dear Reviewer,

Thank you for taking the time to consider our manuscript. Please find our responses uploaded.

Kind regards,

The authors

Reviewer 2 Report

Accessing voluntary HIV testing in the construction industry: A qualitative analysis of employee interviews from the 3 Test@Work study

This paper deals with HIV public health issue and specifically with identification HIV cases at an early stage of disease. The workplace offers a prime place for provision of opt-in HIV testing as part of wider health promotion process. The methodology includes an optional one-off individual health check with tailored health advice and signposting, offered to the construction workforce in health check events delivered as part of the a program named “Test@Work programme”. The participants (N=338) were invited to participate in a semi-structured interview following general health checks which included HIV testing.  Interviewees identified HIV risks including drug use, high numbers of sexual partners and job-related exposures, however, health seeking in construction was limited due to stigma and low support, and nonpermanent workers. The main conclusions were that the organization of the construction industry is complex with multiple organizations of different sizes having responsibility for varying numbers of employees and a discrepancy between organizational policies and employment circumstances is manifested resulted to significant health inequalities. Moreover, the finding resulted to recommending that organizations in the construction sector should offer their employees awareness around health behaviors and protection vial toolbox talks. In addition recommendations are made for annual health checks, sexual health awareness and opt-in workplace HIV testing, aiming to remove barriers to access to healthcare.

This is a qualitative inquiry bases on a large number interviews. Transcripts were analyzed in NVIVO software using an inductive approach, which allowed the coding of the content and the interpretation of themes to be grounded in the participants’ real-life experiences. Even though as qualitative approach a research cannot support validity issues, I think, the use of suitable software and the management of an extensive data collection consists an a positive point of this work.

The paper overall is well written, however there are two methodological issues to be clarified and addressed.

-In the coding procedure, an index estimating the inter-rater (coder) agreement should be reported. This will enhance reliability requirements as far as the coding scheme followed.

- In order to support the findings better, it is suggested, besides the narrative part, to present some networks structures of relations among content elements that consist and represent participants’ perceptions towards the issues in questions. These are kind of concept maps that NVIVO can easily provide.

-Also, sine the interest is limited to HIV, a discussion about generalizing and expended this work to other cases should be more elaborated.  

 Finally, even though the theme is very specific, I think that the paper is worth publishing, after the recommended minor revision.

Author Response

(The authors gave the same response as above.)
